# Short-Form Thymic Stromal Lymphopoietin (sfTSLP) Is the Predominant Isoform Expressed by Gynaecologic Cancers and Promotes Tumour Growth

**DOI:** 10.3390/cancers13050980

**Published:** 2021-02-26

**Authors:** Loucia Kit Ying Chan, Tat San Lau, Kit Ying Chung, Chit Tam, Tak Hong Cheung, So Fan Yim, Jacqueline Ho Sze Lee, Ricky Wai Tak Leung, Jing Qin, Yvonne Yan Yan Or, Kwok Wai Lo, Joseph Kwong

**Affiliations:** 1Department of Obstetrics of Gynaecology, Faculty of Medicine, The Chinese University of Hong Kong, Hong Kong, China; loucia@cuhk.edu.hk (L.K.Y.C.); lautatsan@cuhk.edu.hk (T.S.L.); amychungky@gmail.com (K.Y.C.); kentamc@link.cuhk.edu.hk (C.T.); thcheung@cuhk.edu.hk (T.H.C.); sfyim@cuhk.edu.hk (S.F.Y.); jaclee@cuhk.edu.hk (J.H.S.L.); 2School of Pharmaceutical Sciences (Shenzhen), Sun Yat-Sen University, Shenzhen 510006, China; leungwt@mail.sysu.edu.cn (R.W.T.L.); qinj29@mail.sysu.edu.cn (J.Q.); 3Department of Anatomical and Cellular Pathology, Faculty of Medicine, The Chinese University of Hong Kong, Hong Kong, China; yvonnekin@gmail.com (Y.Y.Y.O.); kwlo@cuhk.edu.hk (K.W.L.); 4School of Medicine, Faculty of Medicine and Health Sciences, Keele University, Newcastle-under-Lyme ST5 5BG, UK

**Keywords:** TSLP, short isoform, ovarian cancer, endometrial cancer, epigenetic regulation, tumour promotion

## Abstract

**Simple Summary:**

Cytokines are a group of small proteins in the body that play an important part in boosting the immune system. Thymic stromal lymphopoietin (TSLP) is a cytokine that plays an important role in the maturation of T cells. Two variants of TSLP, long-form (lfTSLP) and short-form (sfTSLP), have been found, however their roles in cancers are not known. In this study, we discovered that sfTSLP, but not lfTSLP, is predominantly expressed in ovarian and endometrial cancers. The switch that turns the sfTSLP gene on or off is controlled by external modifications of DNA. Our results also found that sfTSLP promotes tumour growth through activating several signal pathways in cancer cells.

**Abstract:**

Thymic stromal lymphopoietin (TSLP) is an epithelial cell derived cytokine belonging to the IL-7 family and a key initiator of allergic inflammation. Two main isoforms of TSLP, classified as long- (lfTSLP) and short-form (sfTSLP), have been reported in human, but their expression patterns and role(s) in cancers are not yet clear. mRNA expression was examined by isoform-specific RT-PCR and RNA in situ hybridisation. Epigenetic regulation was investigated by chromatin immunoprecipitation-PCR and bisulfite sequencing. Tumour progression was investigated by gene overexpression, cell viability assay, cancer organoid culture and transwell invasion. Signals were investigated by proteome profiler protein array and RNA-sequencing. With the use of isoform-specific primers and probes, we uncovered that only sfTSLP was expressed in the cell lines and tumour tissues of human ovarian and endometrial cancers. We also showed the epigenetic regulation of sfTSLP: sfTSLP transcription was regulated by histone acetylation at promoters in ovarian cancer cells, whereas silencing of the sfTSLP transcripts was regulated by promoter DNA methylation in endometrial cancer cells. In vitro study showed that ectopically overexpressing sfTSLP promoted tumour growth but not invasion. Human phosphokinase array application demonstrated that the sfTSLP overexpression activated phosphorylation of multiple intracellular kinases (including GSK3α/β, AMPKα1, p53, AKT1/2, ERK1/2 and Src) in ovarian cancer cells in a context-dependent manner. We further investigated the impact of sfTSLP overexpression on transcriptome by RNA-sequencing and found that EFNB2 and PBX1 were downregulated in ovarian and endometrial cancer cells, suggesting their role in sfTSLP-mediated tumour growth. In conclusion, sfTSLP is predominantly expressed in ovarian and endometrial cancers and promotes tumour growth.

## 1. Introduction

Thymic stromal lymphopoietin (TSLP) is a member of the four-helix-bundle cytokine family and a distant paralog of the cytokine IL-7 [1]. First identified in the supernatant of a mouse thymic stromal cell line, TSLP acts as a growth factor for T and B cells and a co-stimulator for thymocyte proliferation, thus suggesting its role as a lymphopoietin [2,3,4]. The human TSLP homolog was found using in silico methods (identifying the homology to mouse TSLP by a computational screen of human genomic databases; [5,6], which is produced mainly by the epithelial cells of skin, lungs, thymus and intestine [7].)

Both human and mouse TSLPs bind to a heterodimeric receptor consisting of a TSLP receptor chain (TSLPR) and an IL-7 receptor alpha chain (IL-7Rα) [7]. Like IL-7, TSLP activates Signal Transducer and Activator 3 (STAT3) in human and STAT5 in mice and human, and induces the expression of common genes (such as IL-6 and IL-8) [7]. In human, TSLP regulates allergic inflammation at barrier organs such as skin and lungs [8] and plays a key role in their physiopathology including atopic dermatitis and asthma [9,10]. TSLP induces directly the maturation and activation of dendritic cells (DCs) that express the TSLPR/IL-7 heterodimers [11,12], thus promoting an inflammatory T helper type 2 (Th2) response (Th2 cells control immunity to extracellular parasites and all forms of allergic inflammatory responses) [13]. The TSLP-DCs interaction also regulates homeostatic activities in thymuses and intestines, which drives the differentiation/development of regulatory T cells [14,15,16].

Recent studies have been focused on the context-dependent role of TSLP in a wide variety of cancers, which can either promote or inhibit tumour progression [4]. In pancreatic cancer, TSLP is associated with a reduced survival of patients, in which TSLP is produced by tumour-derived IL-1β-activated cancer-associated fibroblasts (CAFs) [17] and favours Th2 cell polarisation via myeloid DC [18]. TSLP produced by keratinocytes in cutaneous T cell lymphoma stimulates tumour growth by inducing Th2 cytokine in the tumour microenvironment [19]. In breast cancer, TSLP produced by cancer cells activates DC and results in Th2 inflammation, which promotes tumorigenesis [20]; whereas in mouse, TSLP can induce immunosuppressive factors by activating CD4+ T cells that promote Th2-mediated immune responses [21]. Studies also showed TSLP enhances the lung metastasis of mammary tumour through an alveolar macrophage-dependent mechanism [22]; and those induced by tumour-derived IL-1α in infiltrating myeloid cells can promote the survival of breast cancer cells and lung metastasis [23]. Nevertheless, the tumour suppressive role of TSLP-mediated inflammation has been reported in the mouse models of breast and skin cancers. TSLP at distant sites of breast cancer leads to robust antitumor immunity by CD4+ Th2 cells [24]. TSLP also causes a severe CD4 and CD8 T cell-dependent Th2-polarised inflammation against skin carcinogenesis [25,26].

Two protein coding transcript variants/isoforms were identified in human bronchial epithelial cells [27], although most studies collectively designated them as “TSLP” as they were not distinguished at mRNA nor protein levels until recently [28]. These two variants, namely TSLP variant 1 (“long-form TSLP”; lfTSLP) and TSLP variant 2 (“short-form TSLP”; sfTSLP), which consist of four and two exons respectively, are derived from the activity of two putative promoter regions [29]. Bjerkan and colleagues reported that the human sfTSLP is a predominant isoform that constitutively expressed at the mRNA and protein level in the keratocytes of oral mucosa and skin and in salivary glands. sfTSLP also exhibits a markedly stronger antibacterial activity than lfTSLP; synthetic sfTSLP does not activate STAT5 signalling in DCs nor interfere with STAT5 activation by lfTSLP [28]. Whilst TSLP isoforms are reported to be responsible for two opposite immune functions [29], both their expressions in the duodenal mucosa of patients with coeliac diseases (CD) were shown to be decreased in active CD mucosa [30]. Nevertheless, lfTSLP contributes to house dust mite (HDM)-induced asthmatic airway epithelial barrier dysfunction, while synthetic sfTSLP prevents this HDM-induced disruption [31]. The roles and differences between lfTSLP and sfTSLP in cancer remain to be determined; here, we have investigated their expression, epigenetic regulation and function in human gynaecologic cancers.

## 2. Materials and Methods

### 2.1. Cell Culture

Five non-malignant immortalised human ovarian surface epithelial cell lines (IOSE20C2, IOSE20C5, IOSE21C21, IOSE25C2 and IOSE25C26; [32]), 2 non-malignant immortalised human fallopian tube secretory epithelial cell lines (FTSEC190 and FTSEC194; [33]), 16 human ovarian cancer cell lines (IGROV-1, PEO1, SKOV3, TOV21G, TOV112D, CaOV4, COV318, COV362, KURAMOCHI, OVKATE, OVSAHO, SNU119, SNU251, TYK-nu, UWB1.289 and A2780), 6 human endometrial cancer cell lines (AN3CA, HEC1A, HEC1B, KLE, Ishikawa and RL95-2), 4 human cervical cancer cell lines (C4-1, Caski, HeLa and HT3) and 1 immortalised human myometrial stromal cell line (hTERT-HM; [34]) were used according to culture methods provided from ATCC or indicated references. Cells were maintained in a humidified incubator at 37 °C with 5% CO_2_.

### 2.2. Patients and Specimens

Fifteen tumour tissues of epithelial ovarian cancer (EOC; including high-grade serous, mucinous, clear cell and endometroid subtypes), 23 tumour tissues of endometrioid endometrial cancer (EEC), 5 tumour tissues of vaginal cancer and 5 tissues of lymph node metastases of ovarian cancer were recruited in this study. Study subjects (patients with ovarian cancer, endometrial cancer or vaginal cancer) were recruited between December 2001 and February 2020 at the Department of Obstetrics and Gynaecology, Prince of Wales Hospital, Hong Kong. Informed written consent was obtained from each patient prior to surgery. The research protocol was approval by the institution’s Clinical Research Ethics Committee and the study was conducted in accordance with the Declaration of Helsinki. Part of the surgical specimens was snapped frozen in OCT compound for frozen tissue sectioning, and part of the specimen was prepared for formalin-fixed paraffin embedded (FFPE) tissue sectioning.

### 2.3. Drug Treatments In Vitro

TSLP-negative ovarian cancer cell lines (A2780, KURAMOCHI and TOV21G) and endometrial cancer cell lines (HEC1A, RL95-2 and KLE) were treated with DNA demethylating agent (5-aza-2′-deoxycytidine; 1 μM; Sigma-Aldrich, St. Louis, MO, USA), HDAC inhibitor (Trichostatin A; 300 nM; Sigma-Aldrich) or histone methyltransferase inhibitor (BIX01294; 5 μM; Sigma-Aldrich) for 72 h. DMSO solvent served as vehicle controls. Genomic DNA and total RNA of the cells after the drug treatments were isolated by AllPrep DNA/RNA mini kit (Qiagen, Antwerp, Belgium).

### 2.4. RT-PCR

Total RNA was extracted from the frozen tissue sections by AllPrep DNA/RNA mini kit or TRIzol Reagent (Invitrogen, Carlsbad, CA, USA), and transcribed into cDNA by high-capacity cDNA kit (Applied Biosystems, Foster City, CA, USA). PCR was performed by Platinum Taq DNA polymerase (Invitrogen) for 35 cycles. Primer sequences for TSLPv1, TSLPv2, TSLPv3 and β-actin (ACTB) were used as follows: TSLPv1 forward 5′-CTC TGG AGC ATC AGG GAG AC-3′ and TSLPv1 reverse 5′-AGT GCT CAA GCA CCA GGT TT-3′; TSLPv2 forward 5′-CCT TGC TCT ACT CAA CCC TGA-3′ and TSLPv2 reverse 5′-CGA GAA AAG GAG AAA ACA CCA-3′; TSLPv3 forward 5′-TCA TCT CAG CAA CCT GAT CG-3′ and TSLPv3 reverse 5′-AGA GCG CTT AAA AGC ACA GC-3′; ACTB forward 5′-CGC GAG AAG ATG ACC CAG AT-3′ and ACTB reverse 5′-GTA CGG CCA GAG GCG TAC AG-3′. Primer sequences for coding region (CDS) of TSLPv1 were used as follows: TSLPv1 CDS forward 5′-GGA ATT GGG TGT CCA CGT AT-3′ and TSLPv1 CDS reverse 5′-TGA CCA TAA TAA AGA TGG TTT ACT GTT-3′ (Appendix A). Primer sequences for lfTSLP, sfTSLP and TSLPR were also applied according to previous reports [29,30],: lfTSLP forward 5′-CAC CGT CTC TTG TAG CAA TCG-3′ and lfTSLP reverse 5′-TAG CCT GGG CAC CAG ATA GC-3′; sfTSLP forward 5′-CCG CCT ATG AGC AGC CAC-3′ and sfTSLP reverse 5′-CCT GAG TAG CAT TTA TCT GAG-3′; TSLPR forward 5′-AGA GCA GCG AGA CGA CAT TC-3′ and TSLPR reverse 5′-CCG GTA CTG AAC CTC ATA GAG G-3′. PCR products were visualised by electrophoresis in 2% agarose gels.

### 2.5. Real-Time Quantitative PCR

Real-time quantitative RT-PCR was performed using Power SYBR Green Master Mix (Applied Biosystems) with TSLPv2-specific primers; and using TaqMan Universal Master Mix II (Applied Biosystems) with TaqMan expression assays for TSLP (Hs00263639_m1), EFNB2 (Hs00187950_m1), PBX1 (Hs00231228_m1), and ACTB endogenous controls by ABI Prism 7900HT Sequence Detection System (Applied Biosystems). Relative mRNA gene expression was calculated by 2^−∆Ct^ method.

### 2.6. Sanger Sequencing

Products of RT-PCR of TSLPv2 were extracted from agarose gel by QIA quick gel extraction kit (Qiagen). The extracted RT-PCR products were cloned to pGEM-T easy vector (Promega, Madison, WI). The pGEM-T vectors with PCR product insert were transformed to DH5a competent *Escherichia coli* (Invitrogen). After bacterial transformation and selection, the pGEM-T vectors with insert were isolated by QuickLyse Miniprep kit (Qiagen). The isolated pGEM vector with insert were subjected to Sanger Sequencing using M13 universal (−43) primer.

### 2.7. RNA In Situ Hybridisation

BaseScope Duplex Assay was performed according to instructions provided by the supplier (Advanced Cell Diagnostics, Newark, CA, USA). FFPE tissues were sectioned at 5 µm thickness on SuperFrost Plus Slides (Fisher Scientific, Waltham, MA, USA) and air-dried overnight at room temperature. Sections were baked at 60 °C for 1 h, deparaffinised in xylene (5 min × 2), 100% ethanol (2 min × 2) and dried at 60 °C for 5 min. Pre-treatment with H_2_O_2_ was applied for 10 min at RT, followed by boiling at 98–102 °C in 1 × Target Retrieval Solution for 15 min. Two rinses in ddH_2_O were performed after each step. Slides were then rinsed with 100% ethanol and dried at 60 °C for 5 min. Protease IV was applied for 30 min at 40 °C and rinsed twice with ddH_2_O. Designed BA-Hs-TSLPv1-2zz-st-C2 probe targeting TSLPv1 (lfTSLP) mRNA (NM_033035.5) and BA-Hs-TSLPv2-3zz-st-C1 probe targeting TSLPv2 (sfTSLP) mRNA (NM_138551.4) were mixed in 1:50 ratio and applied for 2 h at 40 °C. Each sample was probed in parallel with a positive and negative control, presented by housekeeping gene *PPIB/POLR2A* and bacterial *DapB* gene respectively, to evaluate tissue RNA integrity, assay procedure and background signals. Hybridisation with preamplifiers, amplifiers and chromogenic substrates was performed by applying AMP1 (30 min at 40 °C), AMP2 (30 min at 40 °C), AMP3 (15 min at 40 °C), AMP4 (30 min at 40 °C), AMP5 (30 min at 40 °C), AMP6 (15 min at 40 °C), AMP7 (30 min at RT) and AMP8 (15 min at RT), Fast Red substrates (10 min at RT), AMP9 (15 min at 40 °C), AMP10 (15 min at 40 °C), AMP11 (30 min at RT), AMP12 (15 min at RT), and green substrates (10 min at RT). HybEZ oven (Advanced Cell Diagnostics) was employed in all 40 °C incubation. Washing with 1 × Wash Buffer (2 min × 2) was performed between each step. Sections were counterstained with 50% Gill’s Hematoxylin, dried at 60 °C for 15 min, dipped in xylene, and mounted with VectaMount Mounting Medium (Vector Labs, Burlingame, CA, USA).

### 2.8. ChIP-PCR

Chromatin immunoprecipitation (ChIP) was performed in the cell lines with low (IGROV-1 TOV21G, and TOV112D) and high (IOSE20C2 and IOSE25C2) mRNA expression levels of sfTSLP by Simple ChIP Enzymatic IP kit (Cell Signalling, Danvers, MA, USA) using antibodies for acetylated histone H3 and acetylated histone H4 (Cell Signalling). Immunoprecipitated DNA was analysed by qPCR using primer flanking TSLPv2 promoter region. Primer sequences for: TSLPv2 ChIP forward 5′-CAT TTT GGA GAG GGA GTA TCC TG-3′ and TSLPv2 ChIP reverse 5′-CTC CCT AAA TTG GAA CAG AAG TGT-3′.

### 2.9. Bisulfite Genomic Sequencing

Bisulfite conversion was performed on 2 μg genomic DNA using EZ DNA Methylation kit (ZYMO Research, Irvine, CA, USA). The bisulfite converted DNA was amplified by PCR that targeting the TSLPv2 promoter region. Primer sequences for: TSLPv2 CpG forward 5′-TTA GGT ATT TTG GAG AGG GAG TAT TT-3″ and TSLPv2 CpG reverse 5′-AAA TCA AAA TTA AAT AAA ACA AAA AAA A-3′. Target PCR products were purified, cloned to pGEM-T easy vector, and transformed to JM109 High Efficiency Competent Cells (Promega, Madison, WI, USA). Ten clones from each cell line were isolated and sequenced. The percentage of methylation at each CpG dinucleotide (total 29 CpG dinucleotides) in the CpG islands was analysed.

### 2.10. Overexpression of sfTSLP Transcript in TSLP-Negative Cancer Cell Lines

The coding sequencing (CDS) of human TSLPv2 (Appendix A) was cloned by PCR (Platinum Taq DNA Polymerase, Invitrogen) using cDNA derived from a sfTSLP-expressing ovarian cancer cell line (SNU251). Primer sequences for: TSLPv2 CDS cloning forward 5′-GAC GGA TCC CAA AGA AAT GTT CGC CAT GA-3′ and TSLPv2 CDS cloning reverse 5′-CCA GCT AGC TGG TTT ACT GTT GTT TCA GTA AAG GT-3. PCR product was cloned into pZERO-mcs expression vector (InvivoGen, San Diego, CA, USA) using restriction enzymes *BamH*I and *Nhe*I (New England BioLabs, Ipswich, MA, USA). The sfTSLP CDS-expressing vector (named as pZERO-sfTSLP) was confirmed by Sanger sequencing and no mutation was found. The pZERO-sfTSLP was transfected into the TSLP-negative cancer cell lines (A2780, IGROV-1, and HEC1A) by Lipofectamine 2000 reagent (Invitrogen). Transfection of empty plasmid (pZERO-mcs) served as negative controls. Transfected cells were selected by puromycin (InvivoGen). Overexpression of sfTSLP transcript in stable clones was confirmed by qRT-PCR using TaqMan expression assays for TSLP (Hs00263639_m1) since our TSLPv2-specific primers did not cover the CDS of TSLPv2 (Appendix A).

### 2.11. Cell Proliferation Assay

Cell viability was measured by CellTiter-Blue^®^ Cell Viability Assay (Promega, Madison, WI, USA) according to the manufacturer’s instruction. Briefly, 1000 cancer cells (sfTSLP-overexpressing vs. empty vector-expressing) per well were seeded in 96-well plates and cultured in culture medium for 24, 48 and 72 h. After the indicated incubation, the cells were incubated in culture medium containing CellTiter-Blue^®^ Reagent (1:5) at 37 °C for 4–6 h. Cell viability was determined by recording the absorbance at 595 nm.

### 2.12. Tumour Growth Assay by Cancer Organoid Culture

Matrigel (BD Biosciences, San Jose, CA, USA) was loaded into each well of 8-well chamber slide (Nalge Nunc International, Naperville, IL, USA). After gel solidification, cancer cells (sfTSLP-overexpressing vs. empty vector-expressing) were seeded on top of the Matrigel with a density of 10,000 cells per well and incubated for several days. Images of cancer organoids formed in Matrigel were captured by phase-contrast light microscope (100× magnification) after a few days of incubation. Area of cancer organoids in the captured images were analysed by Adobe Photoshop CS4 software (Adobe, San Jose, CA, USA).

### 2.13. Transwell Invasion Assay

Invasive potential of cancer cells was evaluated by Growth Factor Reduced Corning Matrigel Invasion Chamber (Corning, Bedford, MA, USA). Cancer cell suspension (5 × 10^4^; sfTSLP-overexpressing vs. empty vector-expressing) in serum-free culture medium were placed in upper inserts with matrix and membrane. Culture medium with 10% of foetal bovine serum was placed at bottom wells of the companion plate. The invasion chambers were incubated for 24–48 h in a humidified tissue culture incubator at 37 °C, 5% CO_2_ atmosphere. The invaded cancer cells that adhering to the bottom surface of the membrane were fixed by methanol and stained by hematoxylin. Cell counting was facilitated by photographing the membrane through microscope with 200× magnification.

### 2.14. Proteome Profiler Array

Phosphorylation levels of 43 kinase phosphorylation sites and 2 related total proteins in the sfTSLP-overexpressing cells were compared to those of empty vector-expressing cancer cells (A2780, IGROV-1 and HEC1A) by Human Phospho-Kinase Array kit (R&D Systems, Minneapolis, MN, USA). In brief, capture and control antibodies have been spotted in duplicate on nitrocellulose membranes. Cancer cells (sfTSLP-overexpressing vs. empty vector-expressing) were lysed by lysis buffer that provided by the kit. Cell lysates were diluted and incubated overnight with the Human Phospho-Kinase Array. The array was washed to remove unbound proteins followed by incubation with a cocktail of biotinylated detection antibodies. Streptavidin-HRP and chemiluminescent detection reagents were applied. A signal was produced at each capture spot corresponding to the amount of phosphorylated protein bound.

### 2.15. Western Blot Analysis

Amount of 5 μg extracted protein was used for Western blot analysis. Specific primary antibodies were used to probe the target protein: Antibody for TSLP (Abcam, Cambridge, UK, ab47943 and Sigma, St. Louis, Mo, USA, PRS4023), phosphorylated-ERK1/2, ERK1/2, phosphorylated AKT1/2, AKT1/2, phosphorylated-Src, Src, phosphorylated GSK3α/β, GSK3α/β, phosphorylated STAT2, STAT2, phosphorylated p53, p53 (all from R&D Systems), phosphorylated AMPKα1 (Cell Signalling), AMPKα1 (Cell Signalling), EFNB2 (Abcam), PBX1 (Abcam) or β-actin (Sigma-Aldrich). After probing with HRP-linked secondary antibody (GE Healthcare, Chicago, IL, USA), the membrane was detected with ECL reagent (PerkinElmer, Boston, MA, USA).

### 2.16. Immunohistochemistry (IHC)

Sections of 5 μm thickness from formalin-fixed paraffin-embedded ovarian or endometrial tumour specimens were prepared. The slides were deparaffinised, rehydrated in graded ethanol and fixed in neutral buffered formalin. Protein expression of TSLP in ovarian and endometrial tumour tissues was examined by routine IHC procedures using rabbit polyclonal anti-TSLP antibodies (Abcam ab47943 in 1:100 dilution and Sigma PRS4023 in 1:1200 dilution).

### 2.17. RNA Sequencing and Data Analysis

RNA was extracted and purified with the miRNeasy Mini Kit (Qiagen) according to the manufacturer’s protocol. Poly(A)^+^ mRNA was then isolated with the NEBNext Poly(A) mRNA Magnetic Isolation Module (New England Biolabs) as described by the manufacturer. Strand-specific RNA-seq libraries were prepared with the NEBNext Ultra Directional RNA Library Prep Kit (New England Biolabs) following the instructions provided by the manufacturer and sequenced with an Illumina HiSeq 2500 system using the 2 × 150-bp paired-end read mode. For each sample, three independent biological replicates were used for RNA-seq. The procedure of RNA sequencing and data analysis was performed by Novogene (Beijing, China).

### 2.18. Statistical Analysis

All data were described as mean ± S.E.M. and analysed using GraphPad Prism 5.0 software (GraphPad Inc., San Diego, CA, USA) from at least three independent experiments. Statistical difference between experimental groups was determined by Student’s t test and Mann–Whitney U test. *p* < 0.05 were considered as statistically significant (*), *p* < 0.01 as highly significant (**) and *p* < 0.001 as extremely significant (***).

## 3. Results

### 3.1. Short-(sfTSLP), but Not Long-(lfTSLP) Form TSLP, Was Expressed in the Human Cell Lines of Ovarian, Endometrial and Cervical Cancer

Human TSLP is located at chromosome 5q22.1. The human isoforms of TSLP include three transcript variants (RefSeq database), but only two of them give rise to coding RNAs: TSLP transcript variant 1 (TSLPv1; NM_033035; also named as long-form TSLP (lfTSLP)) and TSLP transcript variant 2 (TSLPv2; NM_138551; also named as short-form TSLP (sfTSLP)); TSLP transcript variant 3 (TSLPv3; NM_045089) is a non-coding mRNA ([29]; Figure 1a).

We first studied the expression patterns of TSLP isoforms in ovarian cancer by examining the TSLP transcript variants with RT-PCR in 5 non-malignant immortalised human ovarian surface epithelial (IOSE) cell lines, 2 non-malignant immortalised human fallopian tube secretory epithelial cell (FTSEC) lines and 16 human ovarian cancer cell lines. Locations of the primers designed specifically for TSLPv1, TSLPv2 and TSLPv3 are shown in Figure 1a and Appendix A. We found that the TSLPv2 (sfTSLP) transcript was significantly expressed in four IOSE (IOSE20C2, IOSE20C5, IOSE25C2 and IOSE25C26) and six human ovarian cancer cell lines (PEO1, TOV112D, COV362, SNU119, SNU251 and UWB1.289), whereas the TSLPv1 (i.e., lfTSLP) and TSLPv3 (i.e., the non-coding RNA) transcripts were not expressed in any of the cell lines (Figure 1b). Real time qRT-PCR with TSLPv2-specific primers using SYBR Green PCR master mix showed TSLPv2 transcripts displayed a similar expression pattern with the corresponding IOSE and ovarian cancer cell lines (Figure 1b,c, upper panel). Supplementarily, TaqMan TSLP expression assays (Hs00263639_m1) targeted all 3 transcript variants (TSLPv1, TSLPv2 and TSLPv3) also resembled the expression patterns (Figure 1c, lower panel), thus indicating TSLPv2 (sfTSLP), but not TSLPv1 (lfTSLP) nor TSLPv3 (non-coding RNA), was expressed in human ovarian cancer cells.

RT-PCR assays using the TSLP primer sets designed by Fornasa et al. and Biancheri et al. [29,30] showed the sfTSLP transcript displayed an expression pattern similar to that with our transcript (TSLPv2) in the corresponding cell lines (compared Figure 1b,d), and was also found to be weakly expressed in two FTSEC cell lines (FTSEC190 and FTSEC194) and three ovarian cancer cell lines (SKOV3, CaOV4, OVSAHO). Besides, the lfTSLP transcript was weakly expressed in IOSE25C2, TOV112D and TYK-nu cell lines (Figure 1d). The discrepancy between the two RT-PCR experiments with different primer designs suggested our primer sequences may be more explicit about the TSLP isoforms (Appendix A).

We also showed the TSLPv2 transcript was strongly expressed in three (AN3CA, HEC1B and Ishikawa) out of the six human endometrial and three (C4-1, Caski and HT3) out of the four human cervical cancer cell lines, whereas TSLPv1 and TSLPv3 were not expressed in any of the cell lines being examined (Figure 1e,f). Supplementarily, TSLP receptor (TSLPR) could only be detected in three ovarian (PEO1, COV362 and TYK-nu), two endometrial (HEC1A, RL-95-2) and two cervical (C4-1 and HT3) cancer cell lines (Figure 1b–f). Our results indicated that sfTSLP, but not lfTSLP, was expressed in the cell lines of gynaecologic cancers.

Upon searching the TSLP tissue expression at The Human Protein Atlas (https://www.proteinatlas.org/ENSG00000145777-TSLP/tissue (accessed on 23 December 2020)), we found that the TSLP RNA expression was detected in vagina and lymph node. We used our primers to examine the mRNA expressions of lfTSLP and sfTSLP in five samples of vaginal cancers and five samples of lymph node metastases of ovarian cancer. RT-PCR showed that lfTSLP was undetectable in both the vaginal cancer tissues and lymph node metastases of ovarian cancer (Appendix A).

In order to confirm our RT-PCR results of lfTSLP, we designed another set of primers to amplify the coding region (CDS) of TSLPv1 (lfTSLP; Appendix A). Our RT-PCR again showed mRNA expression of CDS of TSLPv1 could not be detected in any of the vagina cancer tissues or lymph node metastases of ovarian cancer (Appendix A). We therefore concluded that lfTSLP is not expressed in gynaecologic cancers (including ovarian, endometrial, cervical and vaginal cancers) being examined.

On the contrary, the sfTSLP mRNA were detected in all vaginal cancer tissues and lymph node metastases of ovarian cancer by our TSLPv2 primers (Appendix A). The RT-PCR product of TSLPv2 were validated by Sanger sequencing (Appendix A). These results confirmed not only the specificity of our expression assays, but also sfTSLP as the predominant isoform of TSLP in gynaecologic cancers.

### 3.2. sfTSLP Was Expressed in Human Epithelial Ovarian Cancer and Endometrioid Endometrial Cancer

We then examined the expression patterns of sfTSLP (i.e., TSLPv2) in human epithelial ovarian cancer (EOC) (n = 15) and endometrioid endometrial cancer (EEC) (n = 23) tissues from the tumour bank at The Chinese University of Hong Kong. Our TSLPv2 transcript was detected in 13% (2/15) of the EOC (Figure 2a) and 9% (2/23) of the EEC samples (Figure 2b). Besides, RNA in situ hybridisation (BaseScope Duplex assay) showed that positive signal of sfTSLP transcript was more frequently detected in cancer cells of the selected tumour samples with high sfTSLP mRNA (Figure 2c and Appendix A). lfTSLP transcript was not detected in any of the selected EOC and EEC tissues (Figure 2c and Appendix A).

### 3.3. Transcription of sfTSLP in Human Ovarian Cancer Was Regulated by Promoter Histone Acetylation

Next, we investigated if the transcriptional regulation of sfTSLP in ovarian cancer is associated with epigenetic modifications. We treated the TSLP-negative human ovarian cancer cell lines (A2780, IGROV-1, KURAMOCHI and TOV21G) with DNA methyltransferase inhibitor (5-aza-2′-deoxycytidine; AZA), histone deacetylase (HDAC) inhibitor (Trichostatin A; TSA) or histone methyltransferase (HMTase) inhibitor (BIX01294). Results showed that after TSA treatment, sfTSLP (TSLPv2) transcription was upregulated significantly in all the ovarian cancer cell lines being examined, whereas no change could be observed in any of the cell lines after the treatment with AZA or BIX01294 (Figure 3a). This suggested histone tail acetylation in the promoter region regulated active endogenous gene expression of sfTSLP in human ovarian cancer cells. The chromatin immunoprecipitation (ChIP)-PCR assay showed a significant enrichment of acetylated H3K9/14 and acetylated H4K5 in the promoter regions (−332 to −184) of the TSLPv2 gene in IOSE cell lines with high sfTSLP levels (IOSE20C2 and IOSE25C2), but no enrichment was detected in the ovarian cancer cells with low/no expression levels of sfTSLP (IGROV-1, TOV21G and TOV112D) (Figure 3b), thus indicating the gene transcription of sfTSLP promoter in ovarian cancer cells was regulated by histone acetylation.

DNA methylation status at the promoter region and exon 1 region of sfTSLP (TSLPv2; chr5:111073007-111073341; UCSC Genome Browser on Human Dec. 2013 (GRCh38/hg38) Assembly) in IOSE and ovarian cancer cells with or without sfTSLP expression was examined by bisulfite genomic sequencing. Our results showed that promoter DNA methylation of TSLPv2 gene was rare in both the sfTSLP-expressing IOSE cells (IOSE25C2) and sfTSLP-negative ovarian cancer cells (A2780, IGROV-1, KURAMOCHI and TOV21G) (Figure 3c,f), suggesting the silencing of sfTSLP was not regulated by promoter DNA methylation in human ovarian cancer cells.

### 3.4. Silencing of sfTSLP in Human Endometrial Cancer Was Regulated by Promoter DNA Methylation

On the other hand, we found that sfTSLP transcription was up-regulated significantly in two TSLP-negative human endometrial cancer cell lines (HEC1A and RL95-2) after the treatment with AZA, but not TSA or BIX01294 (Figure 3d), suggesting promoter DNA methylation regulated active endogenous gene expression of sfTSLP in human endometrial cancer cell lines. Using bisulfite genomic sequencing of CpG/TpG dinucleotide in the CpG island, our results showed high DNA methylation levels at the promoter and exon 1 region of sfTSLP gene were detected in the HEC1A and RL95-2 cancer cell lines with DMSO control treatment (Figure 3e). The percentage of methylation at each CpG dinucleotide (total 29 CpG dinucleotides) in the CpG islands was also analysed in the cancer cell lines with or without AZA treatment, and our statistical analysis showed that the percentages of methylation of each CpG in the CpG island were significantly decreased in the HEC1A and RL95-2 cells with AZA treatment compared to the cell lines with DMSO (Figure 3e). This indicated sfTSLP silencing in human endometrial cancer cells could be regulated by promoter DNA hypermethylation. Although sfTSLP transcription was up-regulated significantly in TSLP-positive (with low expression levels) human endometrial cancer cell (KLE) after AZA treatment, the promoter DNA methylation was rarely detected in KLE cells with the treatment of AZA or DMSO (Figure 3e). Since sfTSLP transcription in KLE cells was also up-regulated significantly after BIX01294 treatment (Figure 3d), our results suggested that sfTSLP transcription of the KLE endometrial cancer cells was regulated by histone methylation but not DNA methylation.

### 3.5. Overexpression of sfTSLP Promoted Tumour Growth of Ovarian and Endometrial Cancers In Vitro

To investigate the functions of sfTSLP in gynaecologic cancers, the coding sequence of sfTSLP was stably expressed in TSLP-negative ovarian (A2780 and IGROV-1) and endometrial (HEC1A) cancer cells by the transfection of sfTSLP-expressing plasmid (pZERO-sfTSLP). The stable overexpression of sfTSLP in the A2780, IGROV-1 and HECIA clones were confirmed by qRT-PCR (Figure 4a). The following experiments compare the impact of sfTSLP between cells stably overexpressed with sfTSLP and those with empty vector controls. The effect of sfTSLP overexpression on tumour growth in vitro was investigated by cell viability and cancer organoid assays. Results showed that the numbers of viable cells of sfTSLP-expressing A2780, IGROV-1 and HEC1A cells were significantly higher than those of the controls (Figure 4b). Similarly, the sizes of sfTSLP-expressing A2780, IGROV-1 and HEC1A organoids were significantly larger than those of the controls (Figure 4c). Our result therefore indicated that the overexpression of sfTSLP promoted tumour growth of ovarian and endometrial cancers in vitro. However, our transwell invasion assays showed that the numbers of invaded cells of sfTSLP-expressing cancer cells (A2780, IGROV-1 and HEC1A) were not statistically significantly different from those of the control (Figure 4d), thus indicating sfTSLP overexpression did not promote tumour invasion of ovarian and endometrial cancers in vitro.

### 3.6. Overexpression of sfTSLP Activated Intracellular Kinases in Ovarian Cancer Cells

To elucidate the signalling pathways of sfTSLP in gynaecologic cancer cells, we used human phosphokinase array to analyse the phosphorylation of 45 intracellular kinases in sfTSLP-expressing ovarian/endometrial cancer cells (A2780, IGROV-1 and HEC1A). Our results showed that sfTSLP overexpression induced the phosphorylation of AMPKα1, GSK3α/β, STAT2 and p53 in A2780 ovarian cancer cells and the phosphorylation of AKT1/2, ERK1/2 and Src in IGROV-1 ovarian cancer cells (Figure 5a). The capability of sfTSLP overexpression to activate AMPKα1, GSK3α/β and p53 in A2780 cells and AKT1/2, ERK1/2 and Src in IGROV-1 cells was also showed by Western blotting (Figure 5b). In order to demonstrate clearly the signals of AMPKα1, GSK3α/β, p53, AKT1/2, ERK1/2 and Src that had been activated in response to sfTSLP overexpression in A2780 and IGROV-1 ovarian cancer cell lines, we measured and analysed the band intensities of the Western blots (Figure 5b). The band intensity of the phosphorylated or total protein was then normalised to that of its corresponding β-actin expression. Our statistical results showed the levels of phosphorylation of AMPKα1, GSK3α/β and p53 were significantly increased in A2780 cells with sfTSLP overexpression compared to those without overexpression (Figure 5c), indicating AMPKα1, GSK3α/β and p53 signalling pathways were activated in A2780 cells in response to sfTSLP overexpression. Our results also showed that the levels of phosphorylation of AKT1/2, ERK1/2 and Src were significantly increased in IGROV-1 cells with sfTSLP overexpression compared to the controls (Figure 5c), indicating AKT1/2, ERK1/2 and Src signalling pathways were activated in IGROV-1 cells in response to sfTSLP overexpression. sfTSLP overexpression did not affect the activities of any of the 45 intracellular kinases in HEC1A endometrial cancer cells (Figure 5a). Our results thus suggested that the impact of sfTSLP overexpression on the activation of intracellular kinases in ovarian cancer cells was context dependent.

### 3.7. Impact of sfTSLP Overexpression on the Transcriptome of Human Gynaecologic Cancer Cells

Next, we performed RNA-seq in the human gynaecologic cancer cell lines (A2780, IGROV-1 and HEC1A) and compared the transcriptome between ovarian/endometrial cancer cells with and without sfTSLP overexpression i.e., stable transfection of pZERO-sfTSLP vs. stable transfection of pZERO-mcs. In A2780 ovarian cancer cells, a total of 470 differential expressed genes (DEGs) were identified. We analysed the 198 up-regulated DEGs with a 2-fold cut off and found that the biological process associated with sfTSLP overexpression could be categorised into gene ontology (GO) terms including “stem cell differentiation” and “angiogenesis”. Analysis of the 272 down-regulated DEGs with the 2-fold cut off categorised the biological process into GO terms including “anterior/posterior pattern specification”, “regionalisation” and “pattern specification process” (Figure 6a). Among the 608 DEGs being identified in IGROV-1 ovarian cancer cells, 421 DEGs were up regulated and 187 were down-regulated. Analysis of the up-regulated DEGs revealed that the biological process associated with sfTSLP overexpression could be categorised into GO terms including “regulation of endothelial cell migration” and “regulation of epithelial cell migration”. Analysis of the down-regulated DEGs categorised the biological process into GO terms including “type I interferon signalling pathway”, “cellular response to type I interferon” and “response to type I interferon” (Figure 6a).

In HEC1A endometrial cancer cells, 1715 DEGs were identified; the analysis of the 714 up-regulated DEGs revealed that biological process associated with sfTSLP overexpression could be categorised into GO terms including “ribosome biogenesis” and “cell differentiation involved in embryonic placenta development”. Analysis of the 1001 down-regulated DEGs revealed that biological process could be categorised into GO terms including “angiogenesis”, “heart morphogenesis” and “regulation of vasculature development” (Figure 6a). To summarise, the differential expressed gene patterns of the three individual gynaecologic cancer cells with sfTSLP overexpression were associated with distinctive biological processes, suggesting the impact of sfTSLP expression on the transcriptome in gynaecologic cancer was context dependent.

### 3.8. EFNB2 and PBX1 Were Commonly Downregulated in Human Gynaecologic Cancer Cells with sfTSLP Overexpression

We also identified recurrent DEGs in the three human gynaecologic cancer cells with sfTSLP overexpression: A total of 11 common DEGs were found between the 470 DEGs of A2780, the 608 DEGs of IGROV-1 and the 1715 DEGs of HEC1A cells (Figure 6a,b). Among these 11 DEGs (including TSLP), two genes (namely EFNB2 and PBX1) were downregulated in all three human gynaecologic cancer cell lines with sfTSLP overexpression (Figure 6c). The downregulation of EFNB2 and PBX1 in the cancer cells (A2780, IGROV-1 and HEC1A) with sfTSLP overexpression was confirmed by qRT-PCR (Figure 6d). mRNA expression of EFNB2 and PBX1 in IOSE, FTSEC and human ovarian cancer cell lines were also analysed by qRT-PCR (Appendix A). Significant negative correlation between TSLP expression and PBX1 expression was found (Appendix A).

## 4. Discussion

We uncovered that the short-form TSLP (sfTSLP) was predominantly expressed by human ovarian and endometrial tumours and overexpressing the sfTSLP in cancer cells resulted in tumour growth in vitro. In our study, we combined the isoform-specific qRT-PCR with RNA in situ hybridisation to examine the expression pattern of sfTSLP transcript in gynaecologic cancer cells and tumour tissues. Antibodies made with recombinant full-length TSLP as an immunogen recognise both the isoforms, as the C-terminals of sfTSLP protein sequence and lfTSLP sequence are completely overlapped that hinder the generation of an antibody specific to sfTSLP. An indirect approach was used previously by removing sfTSLP reactivity from the polyclonal TSLP antibody to detect the sfTSLP protein [28].

Although there is no antibody available to distinguish between the lfTSLP and sfTSLP, we performed Western blotting to examine protein expression of total TSLP in the ovarian (IGROV-1 and A2780)/endometrial (HEC1A) cancer cell lines with or without sfTSLP overexpression using two polyclonal antibodies. The calculated molecular weight of sfTSLP is 7.4 or 7.1 kDa [28], however the representative band of sfTSLP with expected molecular weight was not detected in the cancer cell lines with sfTSLP mRNA overexpression (Appendix A). On the other hand, multiple bands with different molecular weights were repeatedly detected in the ovarian/endometrial cancer cells with or without sfTSLP overexpression (Appendix A). These findings suggested that the specificity of the two polyclonal anti-TSLP antibodies is low.

Besides, we used the two polyclonal anti-TSLP antibodies to examine the protein expression of total TSLP in ovarian/endometrial cancer tissues by immunohistochemistry. Positive signals were detected in all ovarian/endometrial tumour tissues, including those tissues (O-908 and U-1304) with low expression levels of sfTSLP mRNA (Appendix A). This false positive immunochemical signal might be due to the low specificity of the polyclonal antibodies.

Whilst the expression and effects of TSLP in gynaecological diseases have been investigated and reviewed [35], none of these studies have reported specifically on each TSLP isoform but collectively designated the isoforms as “TSLP”. In ovarian cancer, the expression of TSLP mRNA was significantly higher in tumour tissues and cancer cell lines than the normal controls. High TSLP protein expression in ovarian tumour is significantly associated with clinical parameters including age, histological type, FIGO stage, histological differentiation, pelvis involvement and lymphatic metastasis, and TSLP overexpression is correlated with poor prognosis as it is associated with shorter overall and disease-free survival of the patients [36].

TSLP protein in cervical cancer cells is induced by hypoxia; high TSLP expression is an important regulator of cervical cancer progression by recruiting and licencing tumour-associated eosinophils, which promote the growth of cervical cancer cells [37]. Besides, the interaction between cervical cancer cell-derived TSLP and eosinophils regulates IL-18 and VEGF production, resulting in stimulation of the angiogenesis of human umbilical vein epithelial cells [38]. TSLP also promotes tumour proliferation and invasion of cervical cancer cells through down-regulation of miR-132 expression [39].

Although TSLP has not been reported in endometrial (uterine) cancer, studies revealed in endometrial stromal cells, oestrogen stimulates TSLP production that subsequently promotes growth as well as suppresses apoptosis of the endometrial stromal cells [40,41]. TSLP also is expressed in the stroma of endometrioma tissues, and high TSLP concentrations are found in both the serum and peritoneal fluid of women with endometriosis [42], suggesting TSLP is involved in endometriosis.

We studied the epigenetic regulation of sfTSLP in ovarian and endometrial cancer and found that the transcription of sfTSLP in ovarian cancer cells was regulated by histone acetylation at promoter region, whereas the silencing of sfTSLP transcription in majority of the endometrial cancer cells was regulated by promoter DNA methylation. It is reported that a chromatin remodelling associated with IL-17A-mediated IKKα activity and histone H3 acetylation in Lys14 increase the TSLP mRNA transcription in bronchial epithelial cells during chronic obstructive pulmonary disease (COPD) pathogenesis [43]. Besides, promoter demethylation is found to contribute to the TSLP overexpression in skin lesion of patients with atopic dermatitis (AD), and treating keratinocytes with demethylating agent AZA reduces the methylation of TSLP promoter and increases TSLP transcription [44]. Increased DNA methylation of TSLP locus is also associated with pathogenesis of chronic rhinosinusitis patients with nasal polyps [45]. On the other hand, silencing EZH2 histone methyltransferase downregulates the TSLP expression in oesophagus squamous cell carcinoma cells [46], which supports our observation that histone methylation regulated sfTSLP transcription in KLE human endometrial cancer cells.

Our results showed that sfTSLP was expressed in some of the human ovarian, endometrial and cervical cancer cells. However, TSLPR was not always expressed on the same cell lines. These phenomena could be explained by the hypothesis that sfTSLP exhibits their biological function without binding to TSLPR.

It is well known that lfTSLP exerts its biological activities by binding to heterodimer receptor that consists of IL-7Rα and TSLPR, and the TSLP–TSLPR interaction activates STAT5 signalling pathway via Jak 1 and Jak 2 [7,8]. Bjerkan et al. however demonstrated that sfTSLP did not activate STAT5 signal in dendritic cells nor interfere with STAT5 activated by lfTSLP [28]. Fornasa et al. also addressed the biological activity of the sfTSLP in vitro by conditioning sfTSLP with monocytes-derived dendritic cells, which do not express TSLPR when they are not activated. They found that in the presence of sfTSLP, dendritic cells respond more mildly to bacterial infection. IFN-γ production in allogenic bidirectional mixed lymphocyte reactions from healthy donors is dampened after sfTSLP conditioning and increases in the presence of lfTSLP. They also showed that the anti-inflammatory effect of sfTSLP on human monocyte-derived dendritic cells is mediated via molecular mechanisms involving p38 and ERK1/2 phosphorylation rather than STAT3 and STAT5 phosphorylation through the lfTSLP signals [29,47]. Moreover, Biancheri et al. evaluated whether TSLP isoforms were functional in their system by measuring STAT5 in untreated coeliac disease (CD) biopsies incubated with ex vivo with lfTSLP or sfTSLP. They found that lfTSLP significantly enhanced STAT5 phosphorylation ex vivo, whereas sfTSLP did not have any effect on STAT5 phosphorylation [30]. In our study, the levels of phosphorylation of STAT3 (Y705 and S727), STAT5a (Y694) and STAT5b (Y699) in the sfTSLP-overexpressing ovarian/endometrial cancer cells were similar to those in the cancer cells without sfTSLP overexpression (Appendix A), indicating STAT3 and STAT5 are not activated in the sfTSLP-overexpressing ovarian/endometrial cancer cells. These results suggested that TSLPR or STAT3/5 signalling was not involved in sfTSLP signalling in gynaecologic cancers. Based on this information, we hypothesised that sfTSLP does not bind to TSLPR and activates signalling pathways that are different from lfTSLP.

Our transcriptome assay uncovered that Ephrin-B2 (EFNB2) and Pre-B cell leukaemia homobox-1 (PBX1) were the two recurrently downregulated DEGs in sfTSLP-expressing ovarian and endometrial cancer cells. EFNB2 interacts with its receptor EphB4 through cell-to-cell contact. A recent report by Magic et al. indicated that the introduction of wild-type EFNB2 in breast cancer cells in vitro using stable lentiviral infection was associated with lower cell proliferation, migration and motility compared to controls; in clinical breast cancer specimens, EFNB2 expression was associated with longer patient survival [48]. Based on this information, we hypothesised that the overexpression of sfTSLP promotes tumour growth through the downregulation of EFNB2. In our future work, we are planning to overexpress EFNB2 in a transgenic mouse model to confirm whether the downregulation of EFNB2 is responsible for the tumour growth driven by sfTSLP overexpression.

## 5. Conclusions

In summary, we concluded that sfTSLP was predominantly expressed in ovarian and endometrial cancers and promoted tumour growth.

## Figures and Tables

**Figure 1 cancers-13-00980-f001:**
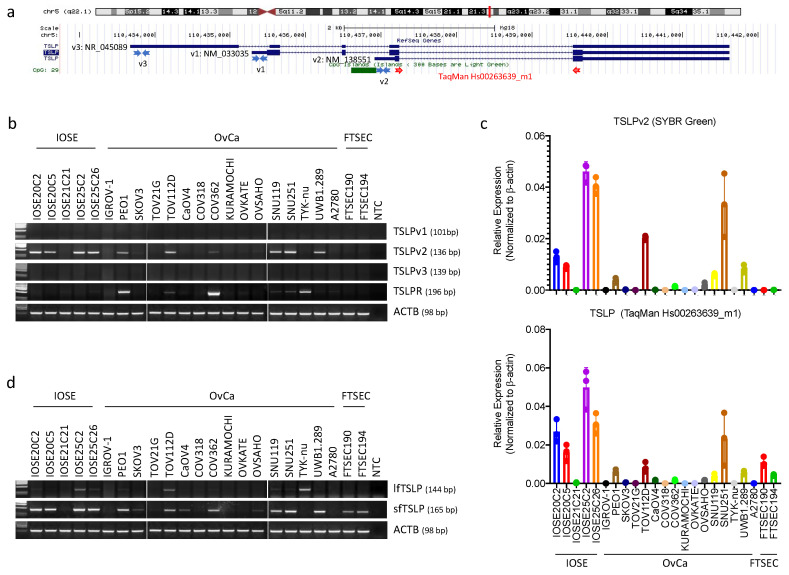
Short-form thymic stromal lymphopoietin (sfTSLP), but not long-form thymic stromal lymphopoietin (lfTSLP), was expressed in the human cell lines of ovarian, endometrial and cervical cancer. (**a**) Schematic representation of the human TSLP locus (5q22.1) from the UCSC Genome Browser (hg18). Primers for RT-PCR of TSLPv1 (NM_033035), TSLPv2 (NM_138551) and TSLPv3 (NR_045089) are labelled as blue arrows; primers and probes of TaqMan expression assay for TSLP (Hs00263639_m1) are labelled as red arrows. A CpG island located at the promoter region and exon 1 of TSLPv2 gene is labelled as green bar. (**b**) RT-PCR of TSLP isoforms (TSLPv1, TSLPv2 and TSLPv3) and TSLPR in 5 non-malignant human immortalised ovarian surface epithelial (IOSE), 2 non-malignant immortalised human fallopian tube secretory tube epithelial (FTSEC) and 16 human ovarian cancer (OvCa) cell lines. β-actin (ACTB) served as house-keeping gene controls. The ACTB controls for IOSE, FTSEC and OvCa cell lines in Figure 1b,d are identical since they are part of the same original RT-PCR experiments with different primer sets of interest. The figures are organized as shown to better align with the data presented in the Result Section; NTC: No template controls. (**c**) Real-time quantitative RT-PCR (qRT-PCR) of TSLPv2 in IOSE, FTSEC and OvCa cell lines was performed by SYBR green assay with TSLPv2-specific primers (upper panel) and qRT-PCR of TSLP was performed by TaqMan expression assay for TSLP (Hs00263639_m1; lower panel). (**d**) RT-PCR of lfTSLP (TSLPv1) and sfTSLP (TSLPv2) in IOSE, FTSEC and OvCa cell lines. Primers specific for lfTSLP and sfTSLP were designed by Fornasa et al. and Biancheri et al. [29,30]. (**e**) mRNA expression of TSLP isoforms (TSLPv1, TSLPv2 and TSLPv3) and TSLPR in 6 human endometrial cancer cell lines (EndoCa) and an immortal human myometrial stromal cell line (hTERT-HM) was examined by RT-PCR (left panel) and qRT-PCR (middle panel: SYBR green assay for TSLPv2; right panel: TaqMan expression assay for TSLP). (**f**) mRNA expression of TSLP isoforms (TSLPv1, TSLPv2 and TSLPv3) and TSLPR in 4 human cervical cancer cell lines (CerxCa) was examined by RT-PCR (left panel) and qRT-PCR.

**Figure 2 cancers-13-00980-f002:**
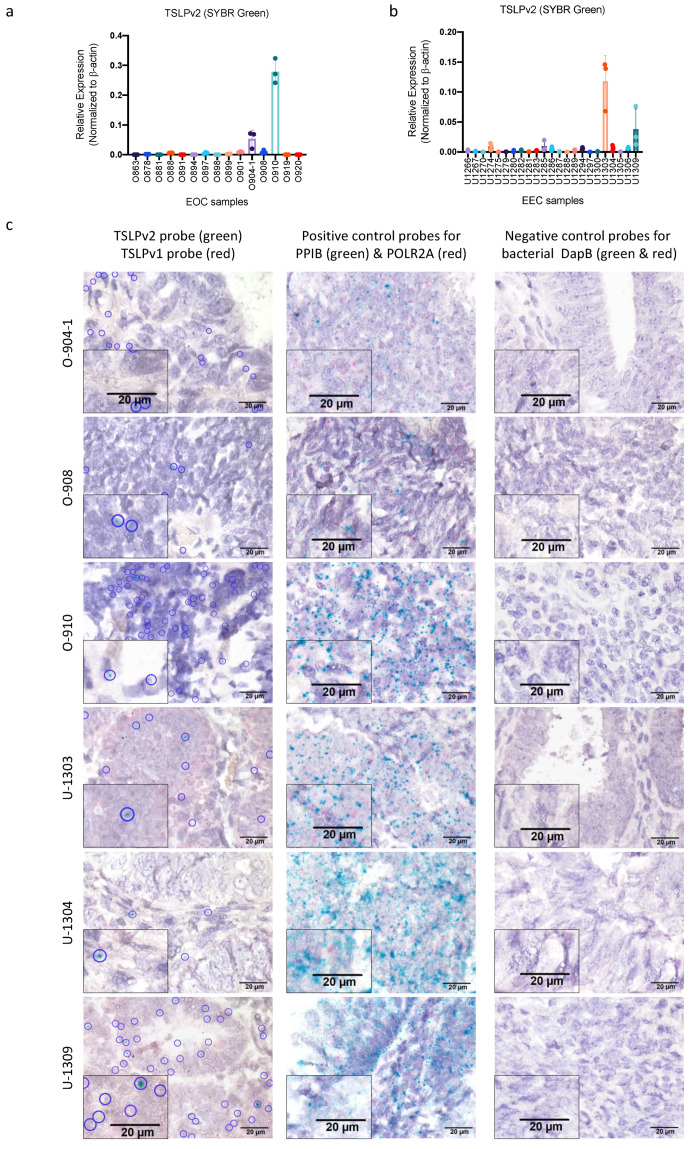
sfTSLP was expressed in human epithelial ovarian cancer and endometrioid endometrial cancer. (**a**) mRNA expression of sfTSLP in 15 tumour tissues of human epithelial ovarian cancer (EOC) was examined by qRT-PCR (SYBR green assay for TSLPv2). (**b**) mRNA expression of sfTSLP in 23 tumour tissues of human endometrioid endometrial cancer (EEC) was examined by qRT-PCR (SYBR green assay for TSLPv2). (**c**) mRNA expression of lfTSLP and sTSLP in selected human EOC and EEC tissues was examined by BaseScope Duplex RNA in situ hybridisation assay. Designed BA-Hs-TSLPv1-2zz-st-C2 (red) targeting TSLPv1 and BA-Hs-TSLPv2-3zz-st-C1 (green) targeting TSLPv2 were applied (left column). Images at 1000× magnification for TSLPv1 and TSLPv2-specific probes (scale bar 20 μm). Blue circles highlighted positive green signals of TSLPv2-specific probe. Positive control probes (for PPIB in green and POLR2A gene in red; middle column) and negative control probes (for bacterial DapB gene in green and red; right column) were also applied. Images at 1000× magnification (scale bar 20 μm). An inserted picture showed higher magnification in of a part of the tissue.

**Figure 3 cancers-13-00980-f003:**
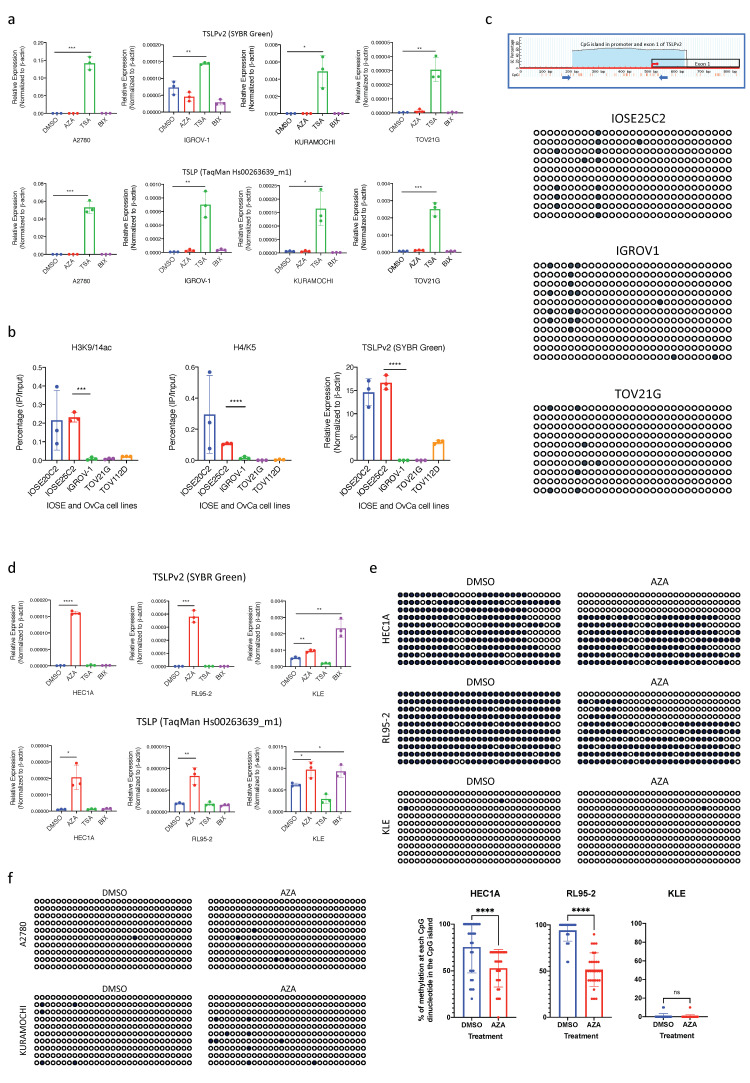
Epigenetic regulation of sfTSLP transcription in human ovarian and endometrial cancer cells. (**a**) mRNA expression of sfTSLP in TSLP-negative human ovarian cancer cell lines (A2780, IGROV-1, KURAMOCHI and TOV21G) after treatment of 5-aza-2′-deoxycytidine (AZA; 1 μM), Trichostatin A (TSA; 300 nM) or BIX01294 (BIX; 5 μM) for 72 h was examined by qRT-PCR (upper panel: SYBR green assay for TSLPv2; bottom panel: TaqMan expression assay for TSLP). DMSO served as vehicle controls. (**b**) Chromatin immunoprecipitation (ChIP) was performed in cell lines with high levels of sfTSLP expression (IOSE20C2, IOSE25C2) and with low/no levels of sfTSLP expression (IGROV-1, TOV21G and TOV112D), using antibodies for acetylated H3K9/14 (H3K9/14ac; left panel) and acetylated H4K5 (H4K5ac; middle panel). Immunoprecipitated DNA was analysed by qPCR using primers flanking promoter region of TSLPv2 (sfTSLP) gene. Expression of sfTSLP transcript in IOSE20C2, IOSE25C2, IGROV-1, TOV21G and TOV112D was shown for reference (right panel, excerpt from Figure 1c). (**c**) CpG content in the promoter region of TSLPv2 gene was analysed using MethPrimer website. A region of genomic DNA (including 500 base pairs upstream of the transcription start site and whole exon 1) was evaluated for the percentage of GC content and individual CpG dinucleotides. Nucleotide position was indicated along the *X*-axis and GC content was graded on the *Y*-axis. A CpG island was found and indicated by shading. Total 29 CpG nucleotides was contained in the CpG island. Each CpG dinucleotide was indicated by a hash mark below the nucleotide numbering, and the transcription start site was indicated by a red arrow. Amplification regions for bisulfite sequencing primers were indicated by blue arrows (upper panel). Bisulfite sequencing of TSLPv2 (sfTSLP) promoter was performed in a sfTSLP-expressing IOSE cell line (IOSE20C2) and two sfTSLP-negative ovarian cancer cell lines (IGROV-1 and TOV21G). ≥10 sequencing were analysed in each cell lines (≥10 rows of the circles; 29 circles in a row); each circle represented a CpG dinucleotide; dark circle represented methylation of the CpG dinucleotide, whereas empty circle represented unmethylation of the CpG dinucleotide. (**d**) mRNA expression of sfTSLP in TSLP-negative/low human endometrial cancer cell lines (HEC1A, RL95-2 and KLE) after treatment of AZA (1 μM), TSA (300 nM) or BIX (5 μM) for 72 h was examined by qRT-PCR (upper panel; SYBR green assay for TSLPv2; bottom panel: TaqMan expression assay for TSLP). DMSO served as vehicle controls. (**e**) Bisulfite sequencing of TSLPv2 (sfTSLP) promoter were performed in sfTSLP-negative endometrial (HEC1A, RL95-2 and KLE). The percentage of methylation at each CpG dinucleotide (total 29 CpG dinucleotides) in the CpG islands was analysed in the cancer cell lines with or without AZA treatment. (**f**) Bisulfite sequencing of TSLPv2 (sfTSLP) promoter were performed in sfTSLP-negative ovarian cancer cell lines (A2780 and KUMRAMOCHI) after treatment of DMSO and AZA (1 μM) for 72 h. Data shown (**a,b,d,e**) are expressed as mean ± S.E.M. from three independent replicates (* *p* < 0.1, ** *p* < 0.01, *** *p* < 0.001, **** *p* < 0.0001).

**Figure 4 cancers-13-00980-f004:**
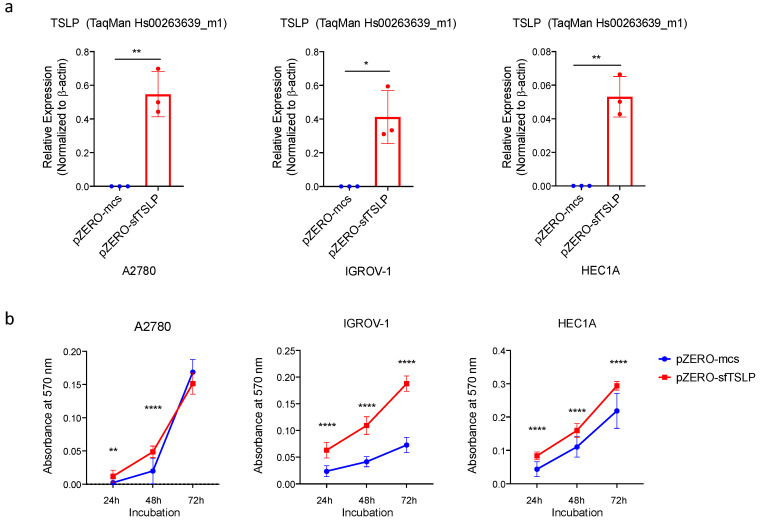
Overexpression of sfTSLP promoted tumour growth of ovarian and endometrial cancers in vitro. (**a**) mRNA expression of sfTSLP in human ovarian (A2780 and IGROV-1) and endometrial (HEC1A) cancer cells with stable transfection of sfTSLP-expression vector (pZERO-sfTSLP) or empty-expression vector (pZERO-mcs) was examined by qRT-PCR (TaqMan expression assay for TSLP (Hs00263639_m1)). (**b**) Cell viability of ovarian (A2780 and IGROV-1) and endometrial (HEC1A) cancer cells with sfTSLP overexpression or empty-vector expression was determined by cell viability assay after 24, 48 and 72 h incubation. (**c**) Cancer organoid formation of ovarian (A2780 and IGROV-1) and endometrial (HEC1A) cancer cells with sfTSLP overexpression or empty-vector expression was captured after a few days of incubation. Scale bar 200 μm. (**d**) Tumour invasion ability of ovarian (A2780 and IGROV-1) and endometrial (HEC1A) cancer cells with sfTSLP overexpression or empty-vector expression was evaluated by transwell invasion assay. Data shown (**a**–**d**) are expressed as mean ± S.E.M. from three independent replicates (* *p* < 0.1, ** *p* < 0.01, **** *p* < 0.0001).

**Figure 5 cancers-13-00980-f005:**
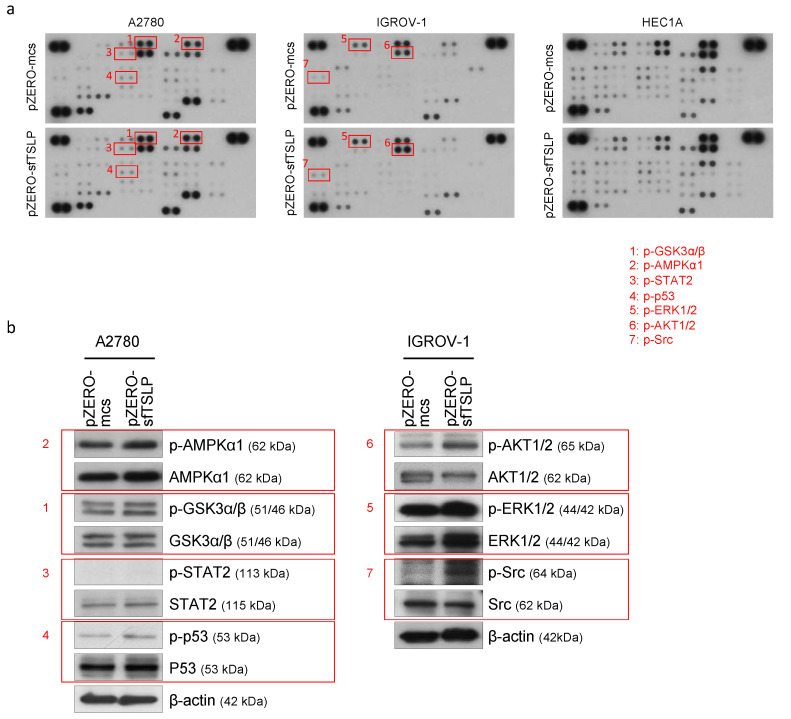
Overexpression of sfTSLP activated intracellular kinases in ovarian cancer cells. (**a**) Human Phospho-Kinase Array was performed in human ovarian (A2780 and IGROV-1) and endometrial (HEC1A) with stable transfection of sfTSLP-expression vector (pZERO-sfTSLP) or empty-vector expression (pZERO-mcs) (upper panel). (**b**) Phosphorylation of GSK3α/β, STAT2 and p53 in A2780 ovarian cancer cells and phosphorylation of AKT1/2, ERK1/2 and Src in IGROV-1 ovarian cancer cells was examined by Western blotting (lower panel). (**c**) To demonstrate clearly the signals of AMPKα1, GSK3α/β, p53, AKT1/2, ERK1/2 and Src that had been activated in response to sfTSLP overexpression in A2780 and IGROV-1 ovarian cancer cell lines, the band intensities of the Western blots were measured and analysed. The band intensity of particular phosphorylated or total protein was normalised to that of their corresponding β-actin expression. The whole Western blots showing all bands and molecular weight markers are included in Appendix A. Data shown (**c**) are expressed as mean ± S.E.M. from three independent replicates (* *p* < 0.1, ** *p* < 0.01, *** *p* < 0.001).

**Figure 6 cancers-13-00980-f006:**
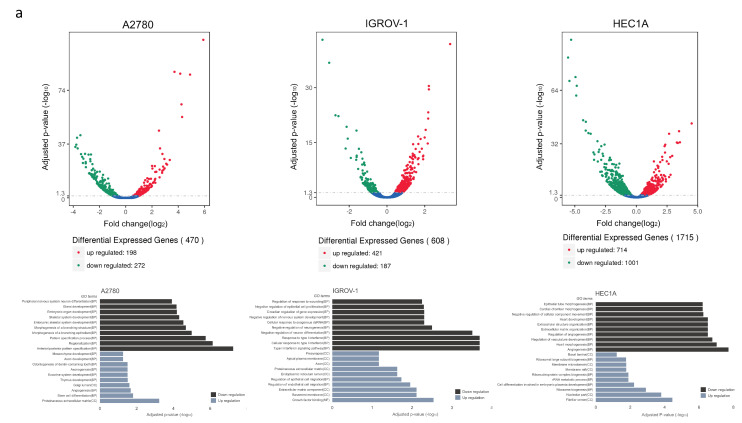
EFNB2 and PBX1 were commonly downregulated in human gynaecologic cancer cells with sfTSLP overexpression. (**a**) The volcano plot showed the upregulated (red dots) and downregulated (green dots) differentially expressed genes (DEGs) of A2780 ovarian cancer cells with sfTSLP overexpression (left top); the top 10 gene ontology (GO) enrichment terms of upregulated DEGs (grey bars) and downregulated DEGs (black bars) were shown (left bottom). The upregulated and downregulated DEGs of IGROV-1 and their top 10 GO enrichment of ovarian cancer cells with sfTSLP overexpression (middle top and bottom). The upregulated and downregulated DEGs and their top 10 GO enrichment of HEC1A endometrial cancer cells with sfTSLP overexpression (right top and bottom). (**b**) The Venn diagram shows relations of DEGs expression by sfTSLP overexpression in A2780 cells (ATvsAM: A2780 cells with stable transfection of pZERO-sfTSLP vs. A2780 cells with stable transfection of pZERO-mcs), IGROV-1 cells (ITvsIM: IGROV-1 cells with stable transfection of pZERO-sfTSLP vs. IGROV-1 cells with stable transfection of pZERO-mcs) and HEC1A cells (HTvsHM: HEC1A cells with stable transfection of pZERO-sfTSLP vs. HEC1A cells with stable transfection of pZERO-mcs). (**c**) Heatmap of the mRNA expression levels of 11 DEGs in three cancer cell lines (A2780, IGROV-1 and HEC1A) with sfTSLP overexpression. (**d**) mRNA expression of TSLPv2, EFNB2 and PBX in ovarian (A2780 and IGROV-1) and endometrial (HEC1A) cancer cells with stable transfection of pZERO-sfTSLP or pZERO-mcs was examined by TaqMan expression assays. Data shown (**d**) are expressed as mean ± S.E.M. from three independent replicates (* *p* < 0.1, ** *p* < 0.01, *** *p* < 0.001, **** *p* < 0.0001).

## Data Availability

The data presented in this study are available on request from the corresponding author. The data are not publicly available due to privacy.

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
