# Peer review of "Short-Form Thymic Stromal Lymphopoietin (sfTSLP) Is the Predominant Isoform Expressed by Gynaecologic Cancers and Promotes Tumour Growth"

_cancers, 2021, doi:10.3390/cancers13050980_

Round 1

Reviewer 1 Report

In this paper the authors examined the expression of thymic stromal lymphopoietin (TSLP) in different gynaecologic cancers (ovarian, endometrial, cervical) trying to identify the short and the long form in their samples, using different techniques. Interestingly they have found that the short but not the long form is predominantly expressed in ovarian and endometrial cancers.

The paper is very interesting but needs more work.

  • In figure 1 the authors present data from RT-PCR. I would like to have an explanation from them about the use of RT-PCR as it is a technique not used anymore to investigate the expression of genes. In particular I cannot understand the usefulness of using this “old” method.
  • Figure 2c: I will suggest to the authors to improve, if possible, the quality of the pictures and maybe to make a bigger magnification of an area of the tissue to see better the positivity in the left column. I cannot distinguish in the middle column the two different positive control. I would like the authors to improve it as well.
  • Figure 3: in Fig 3a they take in consideration the ovarian cancers cell line but not the other cell line negative for TSLP as shown in fig 1: could the authors explain this choice? I think that in all the panel it would be useful to have a positive cell line for TSLP to better appreciate the action of the TSA or AZA or BIX in cell line that do not express TSLP.
  • It would be interesting to understand if their results from RANseq on cell lines overexpressing are confirmed in cell line expressing already TSLP. Could the authors discuss this point? Could they perform qPCR to confirm that the presence of sfTSLP in cell line like IOSE or HEC1B impact their transcriptome? Same for the expression of EFNB2 and PBX1.

Author Response

In this paper the authors examined the expression of thymic stromal lymphopoietin (TSLP) in different gynaecologic cancers (ovarian, endometrial, cervical) trying to identify the short and the long form in their samples, using different techniques. Interestingly they have found that the short but not the long form is predominantly expressed in ovarian and endometrial cancers.

The paper is very interesting but needs more work.

In figure 1 the authors present data from RT-PCR. I would like to have an explanation from them about the use of RT-PCR as it is a technique not used anymore to investigate the expression of genes. In particular I cannot understand the usefulness of using this “old” method.

Response: The purpose of using traditional RT-PCR is to show the cell lines negative for TSLPv1, TSLPv2, TSLPv3 and TSLPR mRNA expression in Figure 1. Since 35 cycles were running in our traditional RT-PCR (Revised in Materials and Methods on p.3 Line 145), the absence of expected PCR product represents the absence of gene expression in the cell lines. On the contrary, it is difficult to show the absence of gene expression in qRT-PCR by undetermined threshold cycle (Ct). In qRT-PCR results, the absence of gene expression presents as a relative low expression levels in delta Ct analysis.

Figure 2c: I will suggest to the authors to improve, if possible, the quality of the pictures and maybe to make a bigger magnification of an area of the tissue to see better the positivity in the left column. I cannot distinguish in the middle column the two different positive control. I would like the authors to improve it as well.

Response: All the images in Figure 2C have been improved by 1000x magnification of an area of the tissue with positive or negative signals (Revised in Results on p.11 Line 434). In the left column, green signals (for TSLPv2 probe) were shown whereas red signal (for TSLPv1 probe) was negative. In the middle column, green positive control (for PPIB probe) and red positive control (for POLR2A probe) were shown. In the right column, two negative controls (for DapB probes) were negative (Revised in Results on p.10).

Figure 3: in Fig 3a they take in consideration the ovarian cancers cell line but not the other cell line negative for TSLP as shown in fig 1: could the authors explain this choice?

Response: There were 11 out of 16 human ovarian cancer cell lines showed downregulation of sfTSLP (Figure 1d). We selected A2780, IGROV-1, KURAMOCHI and TOV21G cell lines for treatment of AZA, TSA or BIX in Figure 3a since they are completely negative for sfTSLP (with undetermined Ct value after 38 cycles of qRT-PCR, Figure 1d).

There were 3 out of 6 human endometrial cancer cell lines showed downregulation of sfTSLP (Figure 1e). We selected HEC1A and RL-95-2 cell lines for treatment of AZA, TSA or BIX in Figure 3d since they are completely negative for sfTSLP (with undetermined Ct value after 38 cycles of qRT-PCR, Figure 1e).

I think that in all the panel it would be useful to have a positive cell line for TSLP to better appreciate the action of the TSA or AZA or BIX in cell line that do not express TSLP.

Response: A human endometrial cell line (KLE) with positive for sfTSLP (with low expression levels; Ct 32 in qRT-PCR at Figure 1e) was used to study the action of AZA, TSA or BIX in Figure 3d (Revised in Results on p.13 Line 509).

It would be interesting to understand if their results from RNAseq on cell lines overexpressing are confirmed in cell line expressing already TSLP. Could the authors discuss this point? Could they perform qPCR to confirm that the presence of sfTSLP in cell line like IOSE or HEC1B impact their transcriptome? Same for the expression of EFNB2 and PBX1.

Response: To examine the expression of EFNB2 and PBX1 in cell lines that expressing already TSLP, mRNA expression of EFNB and PBX1 in IOSE, FTSEC and human ovarian cancer cell lines were analyzed by qRT-PCR (Figure S5a). Associations between TSLP expression and EFNB1 expression, and between TSLP expression and PBX1 expression were analyzed by Spearman’s rank correlation (Figure S5b). Significant negative correlation between TSLP expression and PBX1 expression was found (Revised in Results on p.18 Line 616-619; and revised in Supplementary Materials on p.22 Line 755-758).

Reviewer 2 Report

Interesting study.

It would be of value to follow up in the future the patient-material with evaluations of OS and PFS.

detailed comments are attached.

Author Response

Interesting study.

It would be of value to follow up in the future the patient-material with evaluations of OS and PFS.

detailed comments are attached.

Ref: cancers-1068607

Additional Reviewer comments

Main questions and relevance:

The main questions were; what form of the T-cell promoting cytokine Thymic stromal lymphopoietin (TSLP) is expressed by gynaecologic cancers (mainly ovarian and endometrial), the long-form (lfTSLP) or the short form (sfTSLP) (or both?), and further if actual fTSLP-form promotes tumour growth or not?

The question is highly relevant with the background of cytokines obviously playing a crucial role in activation and promotion of T cell- mounted responses to solid cancers, as just recently shown in the case of muscle invasive urinary bladder cancer (Vollmer et al, Science Translational Medicine, 2021: The intratumoral CXCR3 chemokine system is predictive of chemotherapy response in human bladder cancer | Science Translational Medicine (sciencemag.org)

The intricate balance between T cell driven antitumoral activity versus pro-tumoral activation (growth and/or invasion) has also been shown previously in muscle invasive urinary bladder cancer, in which activation of T regulatory cells in the invasive front of the primary tumor has an end output of downregulating invasion (Winerdal et al: Urinary Bladder Cancer Tregs Suppress MMP2 and Potentially Regulate Invasiveness - PubMed (nih.gov) but on the contrary, in tumor-draining regional lymphnodes has a pro-tumoral end effect (Krantz et al 2018: Neoadjuvant Chemotherapy Reinforces Antitumour T cell Response in Urothelial Urinary Bladder Cancer - PubMed (nih.gov).

Returning to cytokines as active and participating communicators in the immune system, especially between effector cells, APCs and the tumor itself, any investigations of these proteins are of importance. As examplified above , T regulatory cells play different roles in different compartments in the tumor microenvironment and thus different levels of cytokines as well as different isoforms may play different roles (up- or down regulation) and thus investigations are of importance for understanding the minute details.

Originality and Adding novelty pertaining to the subject area

Yes the study is original in clearly showing the important biological connection between a specified cytokine-pathway having an impact on T cell biology in the specified cancers investigated. Thus the study adds novelty in describing and investigating the tumor immunological microenvironment.

Conclusions in the light of the investigations and the results

The conclusions are correct and in accordance with the presented investigations and presented results

Form and style

The form and style is very good.

Response: We appreciate the feedbacks and comments from Reviewer 2.

Reviewer 3 Report

the Authors have addressed an interesting issue and have performed a huge amount of work and they need to be recognized for the effort. However, the paper is too long and mainly observational and presents a long list of molecular biology tests using different techniques (cell lines, human tissue etc.) and their relative results. In fact, the results are mixed with conclusion thoughts that make the paper somehow difficult to read and confusing.

I recognize though that some results are of scientific interest, but they are mixed with other findings that are less significant or sometimes even contradictory and confusing. 

My recommendation is to trim the paper and focus only on cellular in vitro results and possibly confirm the results in a mouse model. Concentrating on one aspect of the biology of TSLP proteins would make the paper more focus and definitely publishable. 

Author Response

The Authors have addressed an interesting issue and have performed a huge amount of work and they need to be recognized for the effort. However, the paper is too long and mainly observational and presents a long list of molecular biology tests using different techniques (cell lines, human tissue etc.) and their relative results. In fact, the results are mixed with conclusion thoughts that make the paper somehow difficult to read and confusing.

Response: The results that are mixed with conclusion thoughts were trimmed. The results of expression of TSLPv1 and TSLPv2 in human breast cancer cell lines were deleted in Figure S3 (revised in Materials and Methods on p.3 Line 117, revised in Results on p.9 Line 366; and revised in Supplementary Materials on p22 Line 753).

I recognize though that some results are of scientific interest, but they are mixed with other findings that are less significant or sometimes even contradictory and confusing. 

Response: The confusing findings were trimmed. The results of protein expression of EFNB1 and PBX1 in ovarian and endometrial cancer cells with or without sfTSLP overexpression were deleted (Revised in Results on p.18 Line 619; revised in Discussion on p.22 Line 739, and revised in Supplementary Materials on p.23 Line 758).

My recommendation is to trim the paper and focus only on cellular in vitro results and possibly confirm the results in a mouse model. Concentrating on one aspect of the biology of TSLP proteins would make the paper more focus and definitely publishable. 

Response: The manuscript was trimmed by deleting irrelevant and confusing results.

Round 2

Reviewer 1 Report

Dear authors thanks for your reply.

I would like you to change your figure 2c because you replied that you have done a higher magnification but the images are exactly the same you just improve the saturation and contrast (much better now). I noticed that on the supplementary data you have done the zoom in of a part of the tissue; it would be better to do in the main figure (you didn't comment this on your reply).

Author Response

Dear authors thanks for your reply.

I would like you to change your figure 2c because you replied that you have done a higher magnification but the images are exactly the same you just improve the saturation and contrast (much better now). I noticed that on the supplementary data you have done the zoom in of a part of the tissue; it would be better to do in the main figure (you didn't comment this on your reply).

Response: Thank you for your comment. Photographs in the left column (RNA probe for TSLPv1 (red) and RNA probe for TSLPv2 (green)) were at 1000x magnification. Images with 1000x magnification were retaken for the middle column (positive control probes) and the right column (negative control probes). All photographs in revised Figure 2c are now at 1000x magnification. Besides, in order to have a better illustration, an inserted picture in each of the photographs in Figures 2c showed higher magnification of a part of the tissue. Revised in Results on p. 10 and on p.11 Line 400.

Reviewer 3 Report

I would like to thank the Authors for reviewing the manuscript and making more concise and focused. I think that the manuscript remains somehow descriptive. some fundamental experiments are lacking which would confirm some of the speculative findings. Overexpressing EFNB2 in a transgenic mouse model could confirm the speculation that EFNB2 downregulation is responsible for some of the findings etc.

I would suggest changing the title to: Short-form thymic stromal lymphopoietin (sfTSLP) is the predominant isoform expressed by gynaecologic cancers AND promotes tumour growth 

Author Response

Reviewer 3

I would like to thank the Authors for reviewing the manuscript and making more concise and focused. I think that the manuscript remains somehow descriptive. some fundamental experiments are lacking which would confirm some of the speculative findings. Overexpressing EFNB2 in a transgenic mouse model could confirm the speculation that EFNB2 downregulation is responsible for some of the findings etc.

I would suggest changing the title to: Short-form thymic stromal lymphopoietin (sfTSLP) is the predominant isoform expressed by gynaecologic cancers AND promotes tumour growth 

Response: Thank you for your comment and suggestion. We agree that some fundamental experiments are still lacking in the study. We therefore are planning to overexpress EFNB2 in a transgenic mouse model in our future study to confirm whether the downregulation of EFNB2 is responsible for the tumour growth driven by sfTSLP overexpression (Revised in Discussion on p. 22 Line 697).

Besides, we fully agree with Review 3 that the title of the manuscript should be changed to “Short-form thymic stromal lymphopoietin (sfTSLP) is the predominant isoform expressed by gynaecologic cancers and promotes tumour growth” (Revised in Title on p. 1 Line 3).

Round 3

Reviewer 1 Report

I want to thank the authors for the proper revision.